# Circulating Multiple Myeloma Cells (CMMCs) as Prognostic and Predictive Markers in Multiple Myeloma and Smouldering MM Patients

**DOI:** 10.3390/cancers16172929

**Published:** 2024-08-23

**Authors:** Ilaria Vigliotta, Vincenza Solli, Silvia Armuzzi, Marina Martello, Andrea Poletti, Barbara Taurisano, Ignazia Pistis, Gaia Mazzocchetti, Enrica Borsi, Lucia Pantani, Giulia Marzocchi, Nicoletta Testoni, Elena Zamagni, Mario Terracciano, Paola Tononi, Marianna Garonzi, Alberto Ferrarini, Nicolò Manaresi, Michele Cavo, Carolina Terragna

**Affiliations:** 1IRCCS Azienda Ospedaliero-Universitaria di Bologna, Istituto di Ematologia “Seràgnoli”, 40138 Bologna, Italy; 2Department of Medical and Surgical Sciences, University of Bologna, 40126 Bologna, Italy; 3Menarini Silicon Biosystems SpA, Via Giuseppe di Vittorio, Castel Maggiore, 40013 Bologna, Italy

**Keywords:** multiple myeloma, smouldering multiple myeloma, liquid biopsy, circulating tumor cells, single-cell analysis

## Abstract

**Simple Summary:**

Although liquid biopsy has emerged as a viable substitute, bone marrow (BM) is still the gold standard for the diagnosis and follow-up of patients with multiple myeloma (MM) and smouldering MM (SMM). The potential involvement of circulating MM cells (CMMCs), counted via CELLSEARCH^®^, in monitoring disease dynamics was assessed by measuring them during treatment and correlating the results with the prognoses of the patients. For MM and SMM patients, the median numbers of CMMCs counted at diagnosis were 349 (1 to 39,940) and 327 (range 22–2463), respectively. Among SMM patients, higher CMMCs were associated with a greater propensity to evolve (*p* = 0.042). The CMMC counts in the MM patients showed a significant correlation (*p* < 0.04) with serum albumin and monoclonal component concentration. Under therapy, CMMCs were consistently detectable in 15/40 patients (coMMstant = 1), and correlated with lower responses (*p* = 0.04) and survival probability (*p* = 0.047), suggesting that CMMC persistence is linked to poor prognoses.

**Abstract:**

In recent years, liquid biopsy has emerged as a promising alternative to the bone marrow (BM) examination, since it is a minimally invasive technique allowing serial monitoring. Circulating multiple myeloma cells (CMMCs) enumerated using CELLSEARCH^®^ were correlated with patients’ prognosis and measured under treatment to assess their role in monitoring disease dynamics. Forty-four MM and seven smouldering MM (SMM) patients were studied. The CMMC medians at diagnosis were 349 (1 to 39,940) and 327 (range 22–2463) for MM and SMM, respectively. In the MM patients, the CMMC count was correlated with serum albumin, calcium, β2-microglobulin, and monoclonal components (*p* < 0.04). Under therapy, the CMMCs were consistently detectable in 15/40 patients (coMMstant = 1) and were undetectable or decreasing in 25/40 patients (coMMstant = 0). High-quality response rates were lower in the coMMstant = 1 group (*p* = 0.04), with a 7.8-fold higher risk of death (*p* = 0.039), suggesting that continuous CMMC release is correlated with poor responses. In four MM patients, a single-cell DNA sequencing analysis on residual CMMCs confirmed the genomic pattern of the aberrations observed in the BM samples, also highlighting the presence of emerging clones. The CMMC kinetics during treatment were used to separate the patients into two subgroups based on the coMMstant index, with different responses and survival probabilities, providing evidence that CMMC persistence is associated with a poor disease course.

## 1. Introduction

Monoclonal gammopathies are hematological neoplasms characterized by the presence of clonal plasma cells (PCs) in the bone marrow (BM). Circulating tumor cells (CTCs) in the peripheral blood (PB), defined here as circulating multiple myeloma cells (CMMCs), can be frequently detected already in the early phases of the disease and, finally, characterize the full stage of plasma cell leukemia [1].

In smouldering (MM) and MM, the definition of a poor outcome (i.e., associated with a high-risk profile) is mainly related to punctual PCs’ cytogenetic characteristics and the out-of-range values of biochemical parameters, such as serum β2-microglobulin (β2M) and lactate dehydrogenase (LDH) [2]. Recently, the establishment of next-generation flow cytometry methods has revealed that a high frequency of CMMCs detected at the time of diagnosis is correlated with poor outcomes, adding CTC enumerations to the already established prognostic factors [3,4,5,6]. However, although the use of novel treatment strategies has significantly improved patient outcomes in both newly diagnosed (ND) and relapsed settings [7], most MM patients continue to relapse. One of the causes might reside in the BM-PCs’ characteristics, particularly in relation to their spreading capacity outside the marrow niche, possibly resulting in the development of CMMC clones and/or clusters with pathogenic features [8,9,10,11].

Therefore, in recent years, the study of CMMCs has aroused much interest as a tool to monitor disease development and/or progression. This includes their potential utility in measuring the minimal residual disease (MRD) as an alternative liquid biopsy method, since the collection of PB is minimally invasive and, consequently, might be frequently repeated during treatment. Moreover, BM aspirates might potentially be subjected to hemodilution affecting residual disease measurements [12,13], while both the enumeration and the characterization of PB-PCs are punctual.

For CMMC detection and isolation, the most commonly used technique is multiparameter flow cytometry (MFC), either by itself or after CD138+ cells’ pre-enrichment. However, since CMMCs are rare and, as the putative presence of uncommon CMMCs should not be excluded, the sensitivity of MFC remains relatively low, requiring a pre-enrichment phase with sufficiently high initial CMMC concentrations [14]. The introduction of next-generation flow cytometry (NGF) offers the opportunity to achieve greater profundity.

Even though it is generally agreed that high CMMC counts are correlated with high-risk features, to date, there is no consensus either on the CMMC cut-off at the time of diagnosis that can be used to identify high-risk patients—also due to the variability of the techniques and detection markers used—or on the efficacy of CMMCs as liquid biopsy alternative analytes to monitor disease progression.

Here, we aimed to analyze CMMCs from the PB of patients affected either by SMM or by MM, using the CELLSEARCH system^®^ (Menarini Silicon Biosystems), a widely-recognized method capable of enriching and enumerating rare circulating cells from PB. CELLSEARCH is approved by the Food and Drug Administration for clinical usage in metastatic breast, prostate, and colorectal cancer [15], and it is considered a sensitive and high-throughput technique for CMMC detection. Thanks to this platform, we intended to establish correlations between CMMC enumeration and patients’ prognoses, and to delineate the CMMCs’ dynamics under treatment. Ultimately, we sought to ascertain whether CMMCs could serve as reliable markers for monitoring disease progression in the context of monoclonal gammopathies.

## 2. Materials and Methods

### 2.1. Patient Cohort

In total, 44 NDMM patients, treated according to daily practice, and 7 SMM patients were consecutively enrolled at the Hematology Institute of IRCCS Azienda Ospedaliero-Universitaria di Bologna, after informed consent was signed. A total of 108 PB (99 from MM and 9 from SMM) samples were collected every 3 months (m) (median follow-up: 6 m, range 0–18 m) throughout their disease course.

In 18/44 NDMM and 7/7 SMM patients, PB samples were also collected at diagnosis, along with BM aspirates. CMMCs were counted from the PB sample within 120 h of sampling by CELLSEARCH system^®^ (Menarini Silicon Biosystems, Bologna, Italy).

Overall, patients had a median age of 61 years (range 46–72) and 59% of them (30/51) were male, as described in Table 1. Most NDMM patients (40/44; 91%) were treated with therapeutic programs including autologous stem cell transplant (ASCT), and half of them (18/44; 48%) were up-front treated with anti-CD38 monoclonal antibodies, in the context of clinical trials and/or of outpatient regimens.

Response to therapy was assessed according to IMWG guidelines [16]; for the purposes of the present study, responses were harmonized with respect to very good partial response (VGPR), with two main categories distinguished: (a) ≥VGPR, and (b) <VGPR.

All clinical information related to NDMM patients included in the study are reported in Appendix A, where the type of therapy provided to patients during induction/first line, transplant (TX) information, and response to therapy are detailed. Briefly, patients were treated with different induction combinations, either including or not including an anti-CD38 treatment: (a) bortezomib, thalidomide, and dexamethasone (VTd), (b) daratumumab–bortezomib–dexamethasone (DARA-VCd), (c) daratumumab with lenalidomide and dexamethasone (DARA-Rd), (d) isatuximab, carfilzomib, lenalidomide, and dexamethasone (ISA-KRd), (e) carfilzomib, lenalidomide, and dexamethasone (KRd), and (f) bortezomib, cyclophosphamide, and dexamethasone (VCd).

### 2.2. BM Sample Manipulation and Characterization

Fresh BM aspirates were collected at diagnosis from both SMM and NDMM patients, to enrich the CD138+ PCs population using anti-CD138 human magnetic micro-beads and AutoMACS^®^ Pro II Separator (Miltenyi Biotec, Bergisch Gladbach, Germany). On the CD138+ enriched fractions, both fluorescence in situ hybridization (FISH) analysis for major MM-related alterations (e.g., t (4;14), del17p, amp1q) and a genomic characterization by ultra-low-pass= whole-genome sequencing (ULP-WGS) to evaluate whole genome’s copy number alterations (CNAs) profile were performed. ULP-WGS (0.1X coverage) was performed on genomic DNA samples from BM-PCs CD138+ cell fractions extracted by Maxwell^®^ (Promega Italia Srl, Milan, Italy) using Maxwell^®^ 16 LEV Blood DNA Kit. Library preparation was completed with SMARTer^®^ ThruPLEX^®^ DNA-Seq kit (Takara Bio, Mountain View, CA, USA). Sequencing was conducted on NextSeq 500 (Illumina Inc, San Diego, CA, USA), and IchorCNA was employed to assess CNAs profiles.

BM cellular immunophenotypes were analyzed via FACSCanto™ II (BD Biosciences, San Jose, CA, USA) using a combination of antibodies provided by BD Biosciences (CD45-V500, CD38-PECy7, CD138-PE, CD19-PerCp Cy5.5, CD56-APC, CD20-APC Cy7, CD81-FITC), by Dako ((Agilent Dako, Santa Clara, CA, USA) lambda-FITC and kappa-APC, by Miltenyi Biotec (CD27-VioBlue (Miltenyi Biotec, Bergisch Gladbach, Germany)), and by BioLegend (CD117-BrilliantViolet 421 (BioLegend, San Diego, CA, USA)). Briefly, 5 µL of each antibody was mixed with 100 µL of fresh BM sample and incubated for 15 min. The sample was then lysed and washed before acquisition. In the analysis tube for cytoplasmatic kappa and lambda chains, a fixation and permeabilization step was also performed. A median of 100,000 events was acquired and no less than 10 events were used to define a cell population. Flow cytometry was also employed to assess the purity of the enriched CD138+ population, with a median of 75% (range 52.4–96.8%).

### 2.3. Minimal Residual Disease (MRD) Assessment

MRD measurements were performed by next-generation sequencing (NGS) after induction therapy and pre-maintenance. Analyses were conducted via LymphoTrack^®^ Dx IgH (FR1/FR2/FR3)/IgK/TCR assays on MiSeq™ System (Illumina Inc., San Diego, CA, USA), with DNA extracted from BM aspirates by Maxwell^®^ (Promega Italia Srl, Milan, Italy) using Maxwell^®^ 16 LEV Blood DNA Kit. MRD measurements were quantified at a sensitivity of at least 10^−5^, using a LymphoQuant B-cell Internal Control. Data analysis was completed by the LymphoTrack^®^ MRD software 2.0.2 (Invivoscribe Inc, San Diego, CA, USA). Undetectable MRD was assessed with a confidence of at least 90%.

### 2.4. CELLSEARCH Enumeration of CMMCs

PC enumerations in PB were performed by employing the CELLSEARCH platform (Menarini Silicon Biosystems) on all 7 SMM patients and 18/44 NDMM patients at diagnosis; conversely, a total of 47 patients (7 SMM and 40 NDMM patients) were monitored every 3 months thereafter. In total, 4 mL per PB sample collected in CellRescue^®^ tubes were processed on CELLSEARCH platform (composed of CellTRACKS^®^ AutoPrep^®^, CellTRACKS Analyzer II^®^, and using CELLSEARCH CMMC assay), as previously reported [17,18]. In brief, CD138+ circulating cells were immune-magnetically enriched with ferrofluid-conjugated anti-CD138+ antibodies, and to differentiate leukocytes from CMMCs, enriched CD138+ cells were stained with CD38-PE, CD19/CD45-APC, and DAPI (to stain nuclei and identify cells). Aberrant CD138+/CD38+/DAPI+/CD19-/CD45- PCs were measured as absolute count per 4 mL of PB. The cartridges containing CMMCs were then stored at 4 °C for subsequent genomic characterizations.

### 2.5. Single-Cell Sorting and Genomic Characterization of CMMCs

Cartridges containing CTCs enriched by CELLSEARCH from 4 MM patients were stored (average storage time = 22 months, range 19–24) and were processed to obtain single cells through DEPArray (Menarini Silicon Biosystems) [19]. Single cells were then whole-genome amplified and quality-controlled using *Ampli1*™ Whole Genome Amplification (WGA) and *Ampli1* QC kits [20], respectively. Low-pass whole-genome sequencing (WGS) libraries were prepared using *Ampli1* LowPass for Illumina (Menarini Silicon Biosystems), according to manufacturer’s instruction, then sequenced on MiSeq (Illumina Inc, San Diego, CA, USA) and analyzed with a custom bioinformatic pipeline (Menarini Silicon Biosystems, based on Ferrarini et al., 2018) [21] to profile CMMCs’ CNAs and determine their ploidy. For each patient, specific yields related to redetection on DEPArray and *Ampli1*™ WGA/ULP-WGS analysis are reported in Appendix A, showing the number of recovered cells analyzed and the number of those passing WGA amplification and sequencing, in-process, quality control (QC) metrics. Overall, 40/125 (32%) of CMMCs enumerated by CELLSEARCH were redetected and isolated as single cells, and 32/40 (80%) of these passed WGA and sequencing quality criteria, thus providing informative results on the cell copy-number profiles, with 30/32 (94%) showing aberrant profiles.

### 2.6. Statistical and Bioinformatic Analyses

Statistical analyses were conducted separately for SMM and NDMM patients, testing both numerical and categorized parameters. The Fisher exact, the Pearson, and the Spearman tests were used to analyze the associations and correlations among variables, respectively. The Kruskal–Wallis tests were used to formally compare medians between groups.

Kaplan–Meier survival curves were obtained via the ggsur-vfit and survival packages to observe the overall trends of progression-free (PFS) and overall survival (OS).

Due to the presence of measurements collected repeatedly over time for the same subjects, a longitudinal regression model was employed [22].

All analyses were performed with R Studio 4.1.2 and the significance level was set to 0.05.

Patients’ BM genomic data were obtained by ULP-WGS of plasma cells. CN profiles were corrected for ploidy using the R package BoBafit 1.8.0 [23].

## 3. Results

Overall, 25 PB samples (eighteen from NDMM and seven from SMM patients) were analyzed by the CELLSEARCH system at diagnosis. The median numbers of CMMCs enumerated at diagnosis were 349 (range 1–39,940) and 327 (range 22–2463) for the NDMM and SMM patients, respectively, showing no difference between the two disease stages (*p* = 0.832, Figure 1A).

### 3.1. CMMC Enumeration in SMM Patients: At Diagnosis and Over Time

Four SMM patients (4/7, 57%) had higher and three (3/7, 43%) had lower CMMC amounts than the median CMMC number counted (*n* = 327). With the limitation of a small sample size, a significant correlation between higher CMMC amount (>327) and PFS (*p* = 0.042, Appendix A) was highlighted, suggesting that patients with higher numbers of CMMCs enumerated at baseline were more likely to evolve rapidly to full MM, as compared to SMM with lower numbers (<327) of circulating cells (13 months vs. 19.5 months, respectively). To date, 4/7 (57%) patients have evolved to MM (AIRC19_13, AIRC19_51, AIRC19_54, and AIRC19_55), with CMMC counts at baseline of 24, 976, 327, and 2463, respectively, and times to progression (TTP) of 34, 19, 2, and 16 months, respectively.

The baseline BM-PC CNAs profiles of the patients with high versus low baseline CMMC amounts were compared, highlighting a prevalence of chromosome 9 amplification (amp9) in the patients with high CMMC amounts (*p* = 0.034), as shown in Figure 1B. Chromosome 9 hyperdiploidies have already been reported to be linked to high-risk SMM [24], supporting the observation that high CMMC counts might be correlated with poor disease outcomes.

The CMMC dynamics were measured in the untreated SMM patients by enumerating them approximately every 3 months, to understand whether a CMMC increment might anticipate progression to MM. The CMMC counts and the major biochemical parameters (e.g., β2M, serum monoclonal component (M-protein), and LDH), as collected at each analyzed time-point, are reported in Appendix A.

Overall, the CMMC dynamics were highly uneven over time and, even in patients who progressed to MM, every measurable CMMC increment was detectable, as shown in Table 2 and in the line-plot depicted in Appendix A. However, in a few patients, the CMMC count was constant throughout the clinical course (e.g., in patient AIRC19_051).

As the SMM patients were not treated, the CMMC dynamics were not conditioned by therapy. Therefore, several other factors, either related to the biological variability of the individual patients or to the circulating cells release normally occurring throughout the day due to circadian rhythms [9,25], should be accounted for to explain the inability to recognize patterns of CMMC dynamics in this cohort of SMM patients.

### 3.2. High CMMC Amounts Describe an Aggressive Phenotype in NDMM Patients at Diagnosis

In 18 NDMM patients, the CMMC counts at baseline were significantly correlated with values of serum albumin (*p* = 0.037), of C-reactive protein (*p* = 0.002), of calcium levels (*p* = 0.004), and of M-protein (*p* = 0.033), whereas a trend was observed with both serum β2M (*p* = 0.061) and BM plasma cell percentage (*p* = 0.07), as shown in Figure 2A, suggesting a more aggressive disease profile in NDMM patients presenting higher CMMC numbers. To eliminate possible discrimination due to outliers, a Spearman’s test was performed, confirming a highly significant correlation with β2M (*p* = 0.008), plasma cell percentage (*p* = 0.035), C-reactive protein (*p* = 0.043), and serum calcium levels (*p* = 0.038), as shown in Figure 3A. This was also confirmed by stratifying the patients in two groups according to the median CMMC number enumerated at diagnosis (*n* = 349): the patients with high CMMC counts (9/18, 50%) had baseline clinical variables defining high risk (the ranges used are shown in Appendix A), such as high serum β2M levels (*p* = 0.044, Figure 1C), high PC percentages in the bone marrow (*p* = 0.04), high calcium (*p* = 0.044) and C-reactive protein (*p* = 0.05, Figure 1D) levels, and ISS III disease stage (*p* = 0.025, Figure 1E).

Overall, 6/18 (33%) patients relapsed, with a median TTP of 10.5 months (range 3–26), with patients with shorter TTP values showing higher baseline CMMC counts (R = 0.93, *p* = 0.021).

In three samples, the CMMCs were also evaluated by flow cytometry, allowing us to compare the CMMC counts to the CMMC frequencies. Briefly, patients AIRC19_035, AIRC19_027, and AIRC19_030 presented 39,940, 18,588, and 316 CMMCs counted with CELLSEARCH at diagnosis, respectively. As for the multiparametric flow cytometry, the patients displayed percentages of 2.5, 0.98, and 0.019, respectively. Once compared simply by proportion, these data resulted in 349 = 0.02%. Notably, this value represents one of the most frequently employed CMMC cut-offs and it is significantly correlated with patients’ prognosis [26,27].

Finally, the BM-PC genomic profiles of the patients with high and low CMMC amounts at baseline were compared, highlighting an over-representation of amp5q in patients with low CMMC counts (*p* = 0.045, Figure 1F), supporting the observation that low CMMC counts might be associated with less aggressive clinical features [28,29]. By contrast, patients with high CMMCs count tend to carry more frequently chromosome 14q deletion (del14q) (*p* = 0.076) and chromosome 1q amplification (amp1q) (*p* = 0.063).

### 3.3. CMMC Counts in MM Patients under Treatment: Comparison with Biochemical Markers’ and MRDs’ Dynamics

In 41 NDMM patients, the CMMCs were enumerated under therapy every 3 months; contextually, the major biochemical parameters were also collected, as shown in Appendix A. Overall, the data were collected in 99 time-points: the CMMCs were enumerated a median of twice per patient within a time-frame including induction and post-ASCT consolidation treatments (i.e., within a median of 6 months from the start of therapy, range 0–18). In this time-frame, a median of 1 CMMC (range 0–5432) was counted in 99 samples analyzed, with a median of 2 CMMCs (range 0–5432) in 52 samples collected under induction therapy, reduced to a median of 0 (range 0–180) in 47 samples, collected under consolidation therapy. Contextually, 2361 biochemical data were measured, as per daily practice, and collected. Local laboratory reference ranges (resumed in Appendix A) were used to define the categorized variables.

By comparing the CMMCs’ and the major biochemical markers’ dynamics, we observed a significant correlation with β2M (*p* < 0.0001), serum albumin (*p* = 0.001), kappa/lambda ratio (*p* < 0.0001), LDH (*p* = 0.042), M-protein (*p* < 0.0001), and total proteins (*p* < 0.0001) (Figure 2B), suggesting that CMMC count might be employed as marker of disease dynamics in NDMM patients under treatment. To avoid possible non-linear data distribution, a Spearman’s rank correlation was also performed, confirming the strong relationship with albumin, total proteins, and M-proteins (*p* < 0.0001) (Figure 3B).

To confirm this observation, a longitudinal regression model was built on the overall amount of data collected over time, including the CMMC counts. To this end, the CMMC counts were considered the dependent variable, whereas β2M, serum albumin, kappa/lambda ratio, LDH, M-protein, and total proteins were identified as the fixed independent variables. The test was corrected for each patient, to mitigate dependence on the intra-subject correlation [22]. The resulting scaled residuals median was extremely low (−0.0532), suggesting the absence of prediction biases. Moreover, despite the high variability between patients, with a variance of 669.17 (standard deviation = 25.87), the model confirmed the highly significant relationship between circulating cells and β2M (*p* = 0.00237), with a 5.5 × increase in the CMMCs amount with every unit increase in the β2M. Even when applying a linear model, a 5.8 × increase in the CMMC amount with every unit increase in β2M was observed (*p*-value= 0.00149), thus further suggesting the inter-relationship between these two variables.

For all the MM patients included in the present study, the MRD was measured in the BM post-induction and post-transplant/before maintenance, to provide benchmark data on the disease dynamics in order to compare with those obtained by liquid biopsy (i.e., CMMC enumeration). MRD was assessed by NGS, detecting the clonal IgH rearrangement(s) defined for each patient at diagnosis. Overall, 42/44 patients were successfully monitored by NGS, whereas 2 were excluded from the analysis, as no clonotypes were identified in their diagnostic sample. Globally, 52 samples at different timepoints (post-induction and post-transplant/before maintenance) were analyzed for MRD. Of these, 28/52 (54%) resulted in an undetectable MRD (sensitivity ≥ 10^−5^). According to the *ALLgorithMM* [30], undetected MRD results measured on hemodiluted samples (10/28 cases, 36%) were excluded, to discount possible false-negative results. Detectable MRDs were found in 24 out of 52 samples (46%), with a median of 1.26 × 10^−3^ residual cells (range 1 × 10^−5^–3.22): in detail, a median of 2.06 × 10^−3^ (range 1 × 10^−5^–3.22) was measured in the post-induction phase (14 measures), while 9.22 × 10^−5^ was the median of the nine measurements assessed during the post-ASCT/pre-maintenance segment (range 1 × 10^−5^–1.29 × 10^−2^).

Overall, CMMCs were undetectable in 18 enumerations out of 42 paired MRD measurements (43%). Of these, 9 cases resulted in an undetectable MRD (9/18, 50%), while 9/18 (50%) were detectable, with a median of 2.52 × 10^−4^ (1 × 10^−5^–9.41 × 10^−3^). Instead, CMMCs were found in 24/42 counts (57%). Paired with the MRD analyses, the median CMMC number was 8 (range 1–619) when the MRD was detectable in 15/24 (median 2.19 × 10^−3^, range 1 × 10^−5^–3.22), whilst in MRD-undetectable cases (9/24, 63%), the CMMCs had a median number of 5 (range 1–19). Even though no significant correlation was highlighted between the residual disease punctual measurements in BM and PB, a tendency between the CMMC and BM MRD values was observed, with 24/42 (57%) concordant paired samples (either undetectable–undetectable or detectable–detectable), as shown in Figure 4 and Appendix A.

### 3.4. A Different CMMCs Dynamic Was Observed in MM Patients after Treatment: The Definition of the coMMstant Index

According to the longitudinal CMMC counts measured in each patient, two possible CMMC dynamic behaviors were observed (Figure 5A): CMMCs were either consistently detectable throughout the disease courses of patients (here, as coMMstant index = 1), irrespective of the treatment provided (Appendix A), or they soon became undetectable, just after the start of treatment (named coMMstant index = 0) (Appendix A). Briefly, the coMMstant = 1 cluster represents patients who consistently showed CMMCs during induction and after ASCT (i.e., CMMCs ≥ 1), while coMMstant = 0 represents MM patients who either did not present CMMCs during follow-up or whose values changed over time, with at least an undetectable count within the first two enumerations.

A coMMstant index = 1 was observed in 15/40 (37.5%) patients, whereas a coMMstant index = 0 was observed in 25/40 (62.5%) patients, where CMMCs were either constantly undetectable or fluctuating over time. Figure 5B reports examples of the two different behaviors.

The high-quality best response rates (≥VGPR and ≥complete response, CR) were significantly lower in the first group, as compared to the second group of patients (6/15 vs. 19/25 ≥ VGPR, *p* = 0.019 and 2/15 vs. 15/25 ≥ CR, *p* = 0.002). Therefore, overall sub-optimal clinical responses were demonstrated by patients with continuously detectable CMMCs under treatment (9/15 vs. 6/25 < VGPR).

Consistently, the BM MRD analyses were positive in 8/10 cases (80%), with coMMstant index = 1, whereas this was true in just 9/23 cases (39%) with coMMstant index = 0 (*p* = 0.05), suggesting that either the persistence of or continuous CMMC release in the peripheral stream is also correlated with sub-optimal molecular responses, as detected in the BM.

Finally, the 18-month PFS values were significantly different in the patients with a coMMstant index = 1, as compared to those of the patients with a coMMstant index = 0 (*p* = 0.047) (Figure 5C), with a hazard ratio (HR) of 4.554, making the risk of relapse 4.5 times higher in the first group of patients, compared to the second group. This translates into a significantly shorter OS in patients with a coMMstant index = 1, as compared to those with a coMMstant index = 0 (p = 0.039, HR = 7.805), as shown in Figure 5D. Moreover, the two different behaviors of the CMMCs dynamics seemed independent from both the amount of CMMCs measured at diagnosis and the risk stratification, considering that the patients with higher CMMCs and/or high-risk stages were equally distributed in the two coMMstant groups.

### 3.5. Single-Cell CMMC CNA Profiles Unveil High Sub-Clonal Heterogeneity

Finally, the CMMCs still detectable after therapy were genomically characterized in four NDMM patients, by sorting single cells by DEPArray after the enumeration and by performing ULP-WGS. To this end, PB samples were collected after a median of 23.5 days of therapy (range 6–70 days); the treatment consisted of a combination of daratumumab (DARA) and bortezomib–ciclofosfamide–desametasone (VCd). The aim of this part of the study was to characterize the CMMCs persisting in the peripheral stream after therapy. For all the patients, the baseline BM-CD138+ whole-genome CNAs profile was also available.

Overall, the genomic analysis of the residual CMMCs showed a substantial overlap with the baseline BM-CD138+ CNAs profiles, as displayed in Figure 6. Indeed, the single-cell analysis added the value of a minute and detailed dissection of the sub-clonal composition, which was barely distinguishable in the genomic profile obtained from the BM-CD138+ bulk analysis (Figure 6A). In addition, the single-cell analysis allowed us to highlight that in three out of four patients, the main ploidy was four (i.e., suggestive of whole-genome doubling events), which again represents a not-so-easily feasible finding in bulk BM-PC characterization.

In detail, the PB from patient AIRC19_031 (CMMC1), collected after 20 days of therapy, had fifteen enumerated CMMCs and, after DEPArray sorting, five single cells were recovered (5/15; 33%) and were evaluable by ULP-WGS. The number of CMMCs with interpretable aberrant CN profiles was three out of five (60%), as described in Appendix A. Overall, the CMMC1′ CNAs profile presented both a private (one out of three cells) gain of chromosome 20q and a ploidy at four (Appendix A), which were not detectable at diagnosis by bulk analysis, probably due to a low tumor fraction (TF, 36%).

In patient number 2 (AIRC19_24; CMMC2), the CMMCs were enumerated from the PB sample after 6 days of treatment: from the initial sixteen circulating cells counted via CELLSEARCH, after 23 months of storage, only two single CMMCs (2/16; 12.5%) were recovered by DEPArray and passed the ULP-WGS analysis quality-control check. The patient, carrying the t (11,14), presented CN variations compatible with the presence of the translocation, considering that one cell showed the amplification of chromosome 11q (amp11q), seen also in the BM-PCs at diagnosis (TF = 42%), and the other cell presented both amp11q and a private deletion of chromosome 14 (Appendix A).

Patient AIRC19_28 (CMMC3) was analyzed after 27 days of therapy, with 32 CMMCs enumerated. After 20 months of storage, 10/32 (31%) were sorted by DEPArray. Six aberrant CMMCs were found after WGA and sequencing QC (6/10, 60%), with a main ploidy fitted at four (i.e., genome doubling event), which was also confirmed by the FISH analysis on the BM-CD138+ cells, but was not detectable by the ULP-WG sequencing of the bulk BM-CD138+ sample, despite the presence of a TF of 79% (Appendix A).

Finally, the data for the last patient (AIRC19_20; CMMC4) were collected and analyzed after 70 days of therapy, showing an amount of 62 residual CMMCs. After 24 months, 23/62 (37%) of these were recovered from the cartridge, with 19 single CMMCs found, with an aberrant CNAs profile, shown in Appendix A. In this last case, the highly heterogeneous BM CNA architecture was disclosed by the single-cell CMMC analysis, and even emerging/new sub-clones were added, present only at the single-cell level after therapy, as shown in Figure 6. A TF of 78% was found on the BM-CD138+ enriched at diagnosis by the ULP-WGS. The genomic profile presented a main ploidy of 1.96, with a highly complex alteration pattern, only partially found in the 19 single cells analyzed: in fact, alterations such as amp/gain1q or del13 were found in 19/19 single cells (100%), alongside del1p, as well as amplifications of chromosomes 6, 9, 18, and 19. However, 13 emerging aberrations (marked with a red box in Figure 6B) were discovered at the single-cell level on residual CMMCs after therapy, suggesting a putative clonal evolution of emerging clones resistant to DARA-VCd and/or a symptom of spatial heterogeneity. In detail, we found three types of new alteration: (a) clonal alterations, present in at least 17/19 cells (89%)—amp11, amp14, amp15, (b) sub-clonal alterations, displayed by at least 12% of cells (>2/19 cells)—amp2, amp16p, del17q, and (c) rare alterations, found in one or, at the, most two cells (<11%)—del3, amp4, amp7q, del14, del15, amp17, del17p, and amp22. Moreover, the bulk BM analysis revealed a telomere focal sub-clonal del8q, not detectable in single cells, where, on the contrary, a clonal whole-arm deletion was present, together with a telomere focal sub-clonal amplification, possibly resulting in a “mixed” event (i.e., one caused by opposite events) that was not detectable via bulk CD138+ analysis due to the summation of single sub-clonal information.

## 4. Discussion

Multiple myeloma (MM) is a highly heterogeneous disease, both from a clinical and from a biological point of view, and this impacts the evaluation of patients’ prognoses and the risk assessments. In clinical practice, bone marrow (BM) aspirate is considered the main source of information for the diagnosis and monitoring of patients with monoclonal gammopathies. Indeed, in recent years, the presence in peripheral blood of circulating elements, such as cell-free DNA, vesicles, microRNAs, exosomes, and circulating tumor cells (CTCs) [31,32,33,34] has emerged as a promising alternative to the conventional bone marrow aspirate, as it is minimally-invasive, although it is informative as to tumor characteristics, and possibly also as to tumor distribution.

Among circulating elements, CTCs have been universally used as disease markers in solid tumors, as they have been proven to mimic disease dynamics and to provide additional information on tumor compositions and settings [35,36,37].

In MM, the release of CTCs has been described for decades [4,14,38,39]. However, it is still unknown which mechanisms drive PC dissemination from BM through PB. Conversely, even though CMMCs and BM tumor PCs generally share similar characteristics [9,40,41], their role as putative markers of the disease’s dynamics is still under study.

With growing knowledge on MM biology and improving technology, CMMCs’ importance in MM biology has been further deepened, and their role as prognostic markers has been hypothesized. In this context, high CTC amount at baseline has been proposed as a marker of high-risk disease; however, neither a consensus on the technique to be used to evaluate their existence nor a CTC count cut-off to consistently identify high-risk MM patients has been established yet [3,4,5,6,14]. Considering the emergent important prognostic role of this peripheral biomarker, the need for validated approaches to detect and, possibly, to finely characterize them is growing. Among the different strategies proposed so far, the CELLSEARCH^®^ system (developed by Menarini Silicon Biosystems) has the advantage of allowing both the absolute enumeration and the collection of CMMCs via a well-established protocol [18,36]. Thus, these cells can be characterized (e.g., by genomic analyses) as bulk or as single cells within a closed system, which enables the recovery and analysis of even a small number of circulating cells (≥10 cells), aiming at the discernment of the complex sub-clonal genomic architecture characterizing plasma cell disorders, which are overall invisible on conventional bulk analysis.

In the present study, we employed the CELLSEARCH system to enumerate the CMMCs in a cohort of newly diagnosed SMM and/or MM patients, both at diagnosis and during the disease course, and we correlated the amount of CMMCs with the prognoses of the patients, highlighting a notable fluctuation in the circulating cell count during the disease progression, which is suggestive of the tumor dynamics under therapy.

With the limitation of the small sample size, we observed that the SMM patients with CMMC counts >327 (i.e., the median of the overall observations in the SMM patients) at least once throughout their disease course, albeit with highly variable CMMCs dynamics, had worse prognoses than the others (*p* = 0.042), suggesting a higher propensity to progression, possibly due to bone marrow tumor burden growth. This was confirmed by a prevalence of chromosome 9 amplification in patients with CMMC counts >327 (*p* = 0.019). In fact, the programmed cell death-1 ligand (PD-L1) gene is located on chromosome 9p24.1, and PD-L1 expression has been reported to be linked to 9 amplification both in SMM and in MM [42], suggesting that SMM patients carrying chromosome 9 amplification might be considered to be more similar to MM than to MGUS [43].

However, as the CMMC dynamics in the SMM patients were mostly uneven, it cannot be excluded that daily oscillations in the circulating cells’ release, due to physiological features and not to disease progression, might have biased the observed results [9,25]. Indeed, even though it was regularly scheduled, the PB collection was actually performed in different moments during the day, and daily oscillations in the CMMC release cannot be excluded, an issue that should be avoided by planning samples’ withdrawal at specific times, in order to focus just on CMMCs dynamics, which are strictly associated with disease progression.

In the NDMM patients, the number of CMMCs enumerated at diagnosis and both the baseline biochemical variables (such as serum beta-2 microglobulin (β2M), serum albumin, C-reactive protein, calcium levels, and serum monoclonal component (M-protein)) and the baseline BM-PC genomic profiles were proven to be correlated (*p* < 0.05), suggesting a more aggressive phenotype in patients with high circulating cell counts, here corresponding to >349 CMMCs (equivalent to 0.02% CMMCs evaluated by flow cytometry). Indeed, high baseline CMMC baseline levels were inversely correlated with chromosome 5 amplifications (amp) (*p* = 0.045), which have already been reported to be a favorable prognostic factor in MM, whereas a trend towards a direct correlation was observed with amp1q (*p* = 0.063) and del14q (*p* = 0.07), two factors commonly associated with a poor prognosis [28,29].

Notably, the MM patients with high baseline CMMC numbers showed lower calcium levels (R = −0.65, *p* = 0.004). Moreover, a trend toward the occurrence of focal lesions (FL > 4, *p* = 0.08) was observed in patients with either a continuous release or a continuous presence of detectable CMMCs, as well as a higher incidence of del17p (*p* = 0.064), a high-risk feature highly frequent in patients with osteolytic lesions [44,45], and a clear presence of del6q (*p* = 0.014), indicating a major risk of progression/aggressive phenotype [46,47]. We might speculate that low serum levels of calcium might act as an attractive signal for CMMCs, causing both their escape from the BM niche and their homing in different skeletal locations, leading to the “first hit” of skeletal lesions, followed by a subsequent rise in serum calcium levels, which was indeed observed in patients just before their progression (as proposed in Appendix A). The hypothesis retraces the CD34+ release from BM found in myelofibrosis (and also in MM patients) [48], driven by an up-take in intracellular calcium and/or calcium-containing matrix vesicle segregation—with consequent low calcium levels—inducing calcium-mediated signals of chemotaxis, here related to CMMC release in the blood stream [49].

In contrast to SMM, the dynamics of the CMMCs under treatment in the MM patients accurately mirrored that of the biochemical indicators, demonstrating highly significant relationships with M-protein, kappa/lambda ratio, and β2M (*p* < 0.0001). Indeed, the strong correlation between β2M and the circulating cells count was also confirmed through the design of a regression model, showing a 5.5-times increase in the CMMC amount with each unit increase in β2M (*p* = 0.00237). Therefore, CMMC dynamics can be considered putative markers of MM disease progression, driven by the selective pressure of treatment.

However, the overall dynamics of CMMCs, rather than their punctual enumerations, were mostly associated with the clinical prognosis of the MM patients. Indeed, two main CMMC patterns under treatment were identified by measuring the CMMCs in at least two successive time-points. In the first one, CMMCS are continuously released (and/or detectable) into the peripheral stream (coMMstant index = 1), whereas in the second one, the CMMC release either ceases completely or presents at least a count = 0 in the first two enumerations (i.e., coMMstant index = 0). This observation allowed us to stratify the patients into two subgroups with different CMMC kinetics, diverse response rates (≥VGPR *p* = 0.019; ≥CR *p* = 0.002), and NGS-determined MRD measurements (*p* = 0.05), supporting the idea that either CMMCs’ persistence in the PB of MM patients is associated with a poor response, or that CMMC clearance represents a favorable prognostic feature. Indeed, this translates into different risks of both progression and death (PFS and OS HR were 4.554 and 7.805, respectively, for patients with coMMstant index = 1, which are significantly higher than those of patients with coMMstant index = 0, *p* < 0.042), and this high risk is associated with the continuous release of CMMCs, irrespective of whether the patient is under treatment. Notably, neither the CMMC counts found at diagnosis nor the ISS or R-ISS classifications were different between the two groups of patients, suggesting that commonly employed prognostic factors might not precisely define patients’ prognoses, unless features descriptive of the disease dynamics are implemented.

Lastly, even in cases when the initial population of CMMCs is limited, the ability to examine the immunophenotypic and genomic profiles of these cells may yield important insights by highlighting high-risk characteristics that may impart aggressive and/or resistant phenotypes. In fact, by comparing the single-cell analysis of the residual CMMCs after therapy to the baseline BM-PCs’ genomic bulk analysis, we have been able to identify sub-clonal alterations that emerge post-therapy, potentially as a result of selective therapeutic pressure or the circulation of clone(s) that may originate from a different niche.

In conclusion, both the intrinsic properties of CMMCs and their continuous release, observed in about 37.5% of MM patients, are more informative than their baseline levels, pointing to a potential function for CMMCs as seeds of therapy resistance and/or of disease dissemination.

Owing to the small patient cohort examined in this analysis and the limited follow-up, these observations serve as a proof of concept, which require additional testing in a larger patient group.

## 5. Conclusions

The present study suggests that baseline CMMC counts alone, both in SMM and in MM, are not sufficiently descriptive of the disease’s dynamics, despite the significant correlations observed between CMMC amounts and clinical–biochemical features. Therefore, repeated monitoring, even in the early phases of the disease course, is needed to precisely define and monitor patients’ outcomes. To this end, peripheral blood collection, which is minimal invasive, seems more appropriate than BM aspiration, even though, to date, BM is considered the gold standard sampling method for the measurement of disease dynamics in monoclonal gammopathies. Moreover, circulating elements might collect information derived from different lesions, thus providing details on the tumor composition and distribution. Ultimately, peculiar prognostic features can be distinguished among patients either continuously releasing or not continuously releasing CMMCs, with more aggressive and resilient phenotypes associated with the first behavior, even in instances when the initial CMMC count is low.

## Figures and Tables

**Figure 1 cancers-16-02929-f001:**
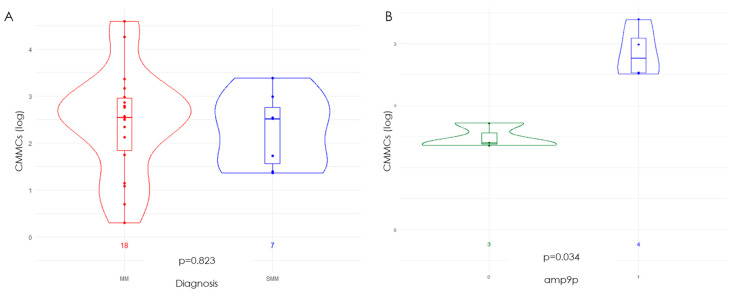
Kruskal–Wallis analyses panel (CMMC count expressed as logarithm value). (**A**) CMMC counts in MM (red) and SMM (blue) patients at diagnosis; (**B**) correlation between CMMC count and amplification (amp) of chromosome 9 in SMM patients, with SMM patients carrying amp9 displayed in blue; (**C**) correlation between CMMC counts in MM patients and serum beta-2 microglobulin (β2M) levels; (**D**) correlation between CMMC counts in MM patients and c-reactive protein (CRP); (**E**) correlation between CMMC counts in MM patients at baseline vs. ISS III (in orange ISS III-patients); (**F**) correlation between CMMC counts in MM patients and chromosome 5q amplification.

**Figure 2 cancers-16-02929-f002:**
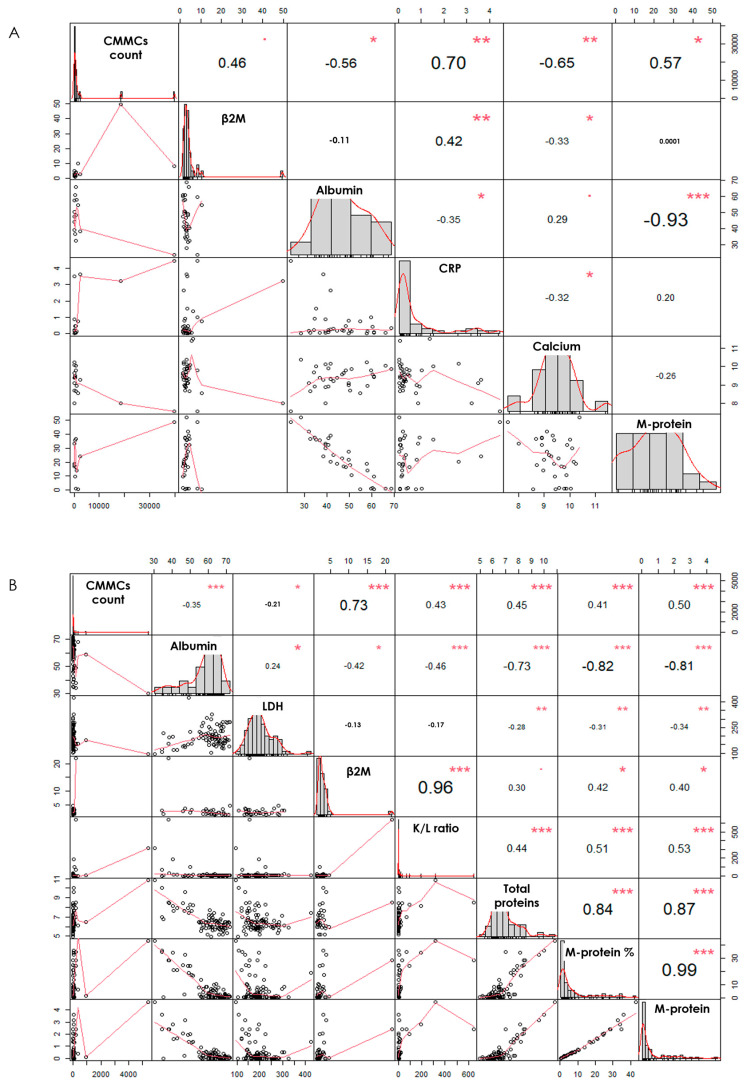
Graphical correlation matrix (Scatterplot matrix, Pearson’s test). The absolute correlation between pairs of variables is displayed in the upper panels, with the font size proportional to the absolute value of the correlation. Statistical significances are highlighted with * (* = 0.01; ** = 0.001; *** = 0.0001). Along the diagonal are presented the histograms for each variable, and the LOESS (locally estimated scatterplot smoothing) curves are displayed in the lower panels. (**A**) Correlation between CMMC count and major biochemical markers’ continuous variables at diagnosis, and (**B**) during post-treatment pre-maintenance phase.

**Figure 3 cancers-16-02929-f003:**
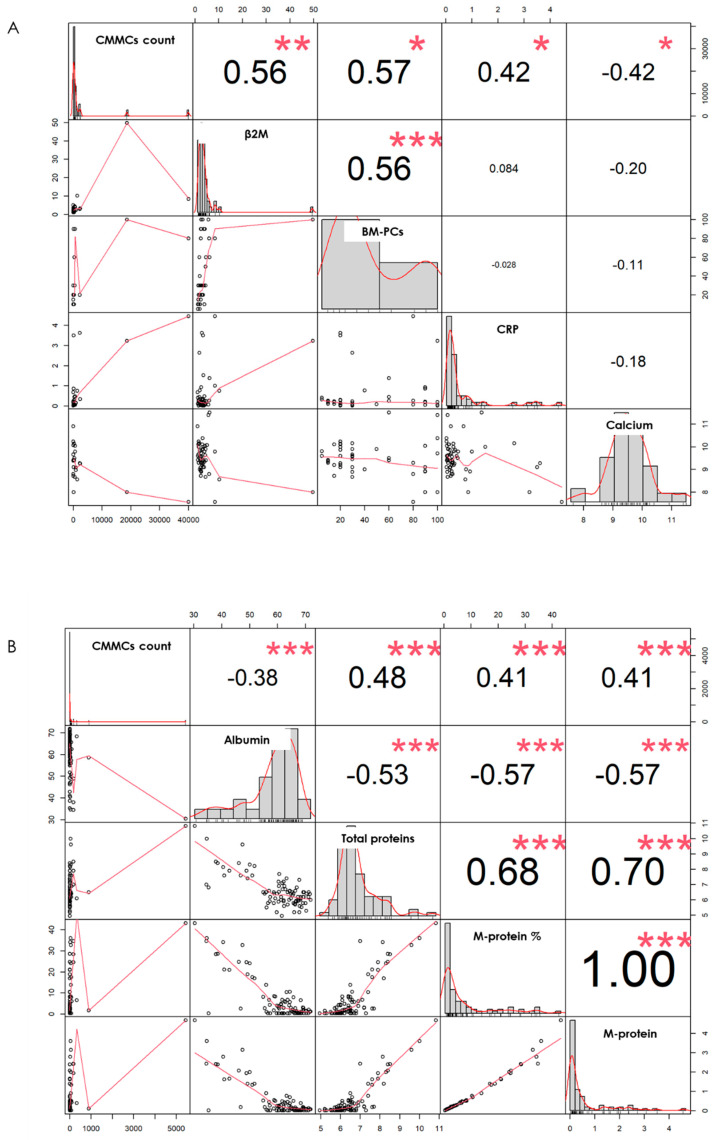
Graphical correlation matrix (Scatterplot matrix, Spearman’s test). The upper panels show the absolute correlation between pairs of variables, with the font size corresponding to the correlation’s absolute value. The histograms for each variable are shown along the diagonal, and the lower panels show the LOESS curves. (**A**) Correlation between CMMC counts and major biochemical markers’ continuous variables at diagnosis, and (**B**) during post-treatment pre-maintenance phase. Statistical significances are highlighted with * (* = 0.01; ** = 0.001; *** = 0.0001).

**Figure 4 cancers-16-02929-f004:**
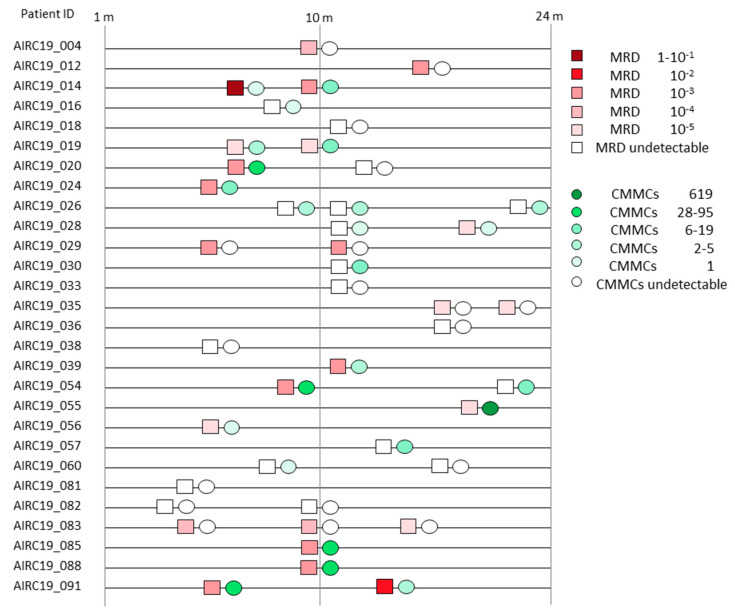
Paired analyses of MRDs and CMMCs (i.e., performed at the same time) at different time-points (m = months) for each MM patient. MRD measures are displayed as squares, while CMMCs enumerations are shown as circles. MRD and CMMC levels are explained in the legend.

**Figure 5 cancers-16-02929-f005:**
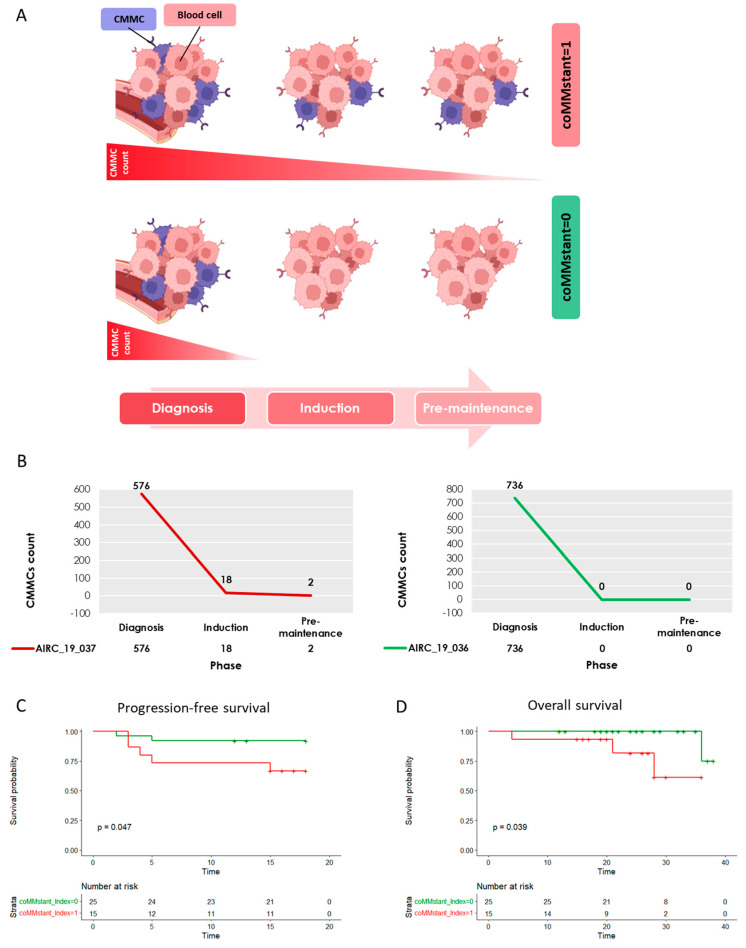
Patient clustering by CMMCs counted throughout treatment monitoring. (**A**) A graphical display of the two different CMMC dynamics in the coMMstant index. Patients were divided into two clusters by their CMMC numbers: coMMstant = 1 (in red) patients, who consistently presented CMMCs during induction and after ASCT, and coMMstant = 0 (in green), who were not characterized by CMMCs during follow-up, or whose number increased or decreased over time, show at least a count = 0 within the first two enumerations. Cell design was performed by BioRender^®^ (https://app.biorender.com/). (**B**) Examples of a coMMstant = 1 patient (in red, right) and a coMMstant = 0 patient (in green, left) with their CMMCs dynamics through disease monitoring: at diagnosis, during induction, and in pre-maintenance. Survival probability curves between coMMstant = 1 and coMMstant = 0 groups: (**C**) Progression-free survival according to coMMstant index (months), and (**D**) overall survival according to coMMstant index (months).

**Figure 6 cancers-16-02929-f006:**
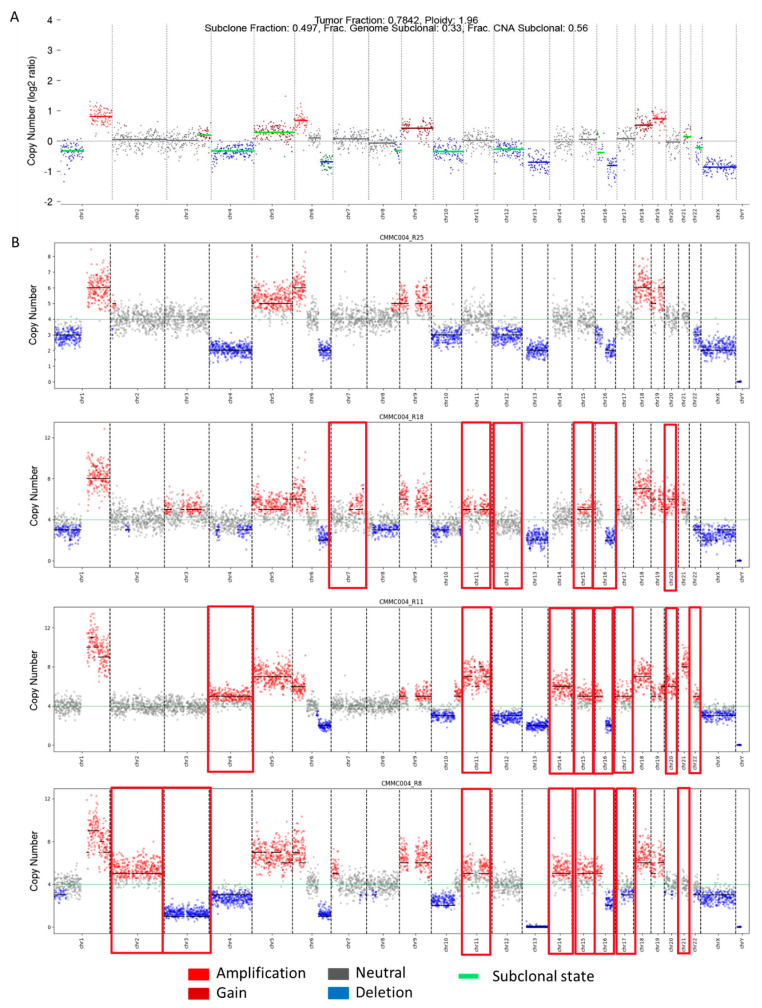
Single-cell genomic analysis of CNA profiles. (**A**) BM-PC CNA profile; (**B**) CMMC CNA profiles of four single cells. In red are highlighted gains and amplifications, and in blue are deletions. Gray symbolizes normal status. CNAs differing from single cells to BM profiles are denoted with a red box.

**Table 1 cancers-16-02929-t001:** Patients overview. The cohort’s main characteristics are listed.

	Median	Range		
**Age (y)**	61	46–72		
	Female (%)	Male (%)		
**Gender**	21 (41)	30 (59)		
	SMM (%)	MM (%)		
**Disease phase**	7 (14)	44 (86)		
	Median SMM	Range SMM	Median MM	Range MM
**BM plasma cells (FC)**	2.4%	1.3–14%	2.7%	0.1–40%
	Kappa (%)	Lambda (%)	Unknown (%)	
**Light chain type**	38 (64)	20 (34)	1 (2)	
	I stage (%)	II stage (%)	III stage (%)	Unknown (%)
**ISS**	26 (51)	14 (27)	5 (10)	6 (12)
	I stage (%)	II stage (%)	III stage (%)	Unknown (%)
**R-ISS**	21 (41)	18 (35)	2 (4)	10 (20)
	Median MM	Range MM	Median SMM	Range SMM
**CMMCs at diagnosis**	349	1–39,940	327	22–2463

y = years; FC = flow cytometry; CMMCs are expressed as absolute count per 4 mL.

**Table 2 cancers-16-02929-t002:** SMM patients’ CMMCs counts, highlighting months from diagnosis and post-treatment in those patients who progressed to MM.

	Patient ID	Disease Phase	Time from Diagnosis (m)	CMMCs Count
**Non-progressive SMM**	AIRC19_001	SMM	17	22
	AIRC19_001	SMM	21	2
	AIRC19_001	SMM	38	52
	AIRC19_021	SMM	2	52
	AIRC19_021	SMM	7	159
	AIRC19_021	SMM	13	49
	AIRC19_021	SMM	26	138
	AIRC19_075	SMM	8	344
	AIRC19_075	SMM	12	656
	AIRC19_075	SMM	16	371
**SMM patients who** **progressed to MM**	AIRC19_013	SMM	12	24
	AIRC19_013	SMM	23	100
	AIRC19_013	MM	1	0
	AIRC19_051	SMM	1	976
	AIRC19_051	MM	0	960
	AIRC19_054	SMM	1	327
	AIRC19_054	MM	0	38
	AIRC19_055	SMM	4	2463
	AIRC19_055	SMM	11	390
	AIRC19_055	MM	0	619

m = months.

## Data Availability

All data generated or analyzed during this study are included in this published article (and its Appendix A).

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
