# Peer review of "Circulating Multiple Myeloma Cells (CMMCs) as Prognostic and Predictive Markers in Multiple Myeloma and Smouldering MM Patients"

_cancers, 2024, doi:10.3390/cancers16172929_

Round 1

Reviewer 1 Report

Comments and Suggestions for Authors

This original article investigates the use of circulating multiple myeloma (MM) cells as biomarkers for MM and smoldering multiple myeloma (SMM) patients. The authors also aim to identify alternative biomarkers due to the challenges and costs associated with enumerating CMMCs in peripheral blood. Furthermore, the study analyzes amplifications in chromosomes 5 and 9, which led the consistency of the previous reports and providing additional evidence for a potential linkage with the programmed cell death-1 ligand (PD-L1) gene. The authors also attempt to determine a threshold number of CMMCs that could serve as a prognostic indicator and measure response to standard therapeutic strategies, establishing the coMMstant index for MM and SMM patients. The overall strategy and dedication of the authors deserve commendation.

However, it is noteworthy that the sample size is slightly smaller than what is typically reported in similar original research articles. Despite this limitation, the study's concepts and the detailed examination of CMMC kinetics are remarkable. This study is particularly relevant to clinicians focused on therapies for incurable diseases. Consequently, I believe this manuscript merits publication in "Cancers" after appropriate revisions and updates based on reviewers’ comments, including those provided herein.

Several points should be addressed to improve the manuscript before publication:

  1. While it may be a matter of preference, a graphical abstract might not be necessary for this article.
  2. The introduction is generally well-stated, but there are several unnecessary hyphens, such as in “dis-ease” (line 51) and “re-quiring” (line 75). These should be removed unless there is a specific reason for their inclusion.
  3. The reference style of [15] on line 88 seems inconsistent with the journal's citation style, which uses square brackets rather than parentheses.
  4. In the results section, it is recommended to describe the results more clearly and systematically. Some readers may not have an advanced understanding of statistical methods, which could hinder their comprehension. Specifically, Figures 2 and 3, generated using R software, differ from typical graphical correlation matrices. Therefore, more detailed figure captions are necessary to guide readers to the relevant results in the text.
  5. Please reconsider the sequence in which the results are presented in the text. The current order appears somewhat disjointed (e.g., Figures 1A, B, Table 2, Figures 2A, 3A, 1C, 1D, 1F, 1E, 2B, 3B, 4, and 5A-D). A more logical and sequential presentation would enhance clarity and coherence.
  6. Finally, the results section should be strictly objective. Phrases such as “suggesting that ~~~”, “Notably, ~~~”, or “Interestingly, ~~~” should be avoided as they introduce subjective interpretation. Results should be stated factually without subjective commentary.

Reviewer 2 Report

Comments and Suggestions for Authors

This is a potentially useful paper with interesting results. It is very difficult to follow some sections and the Figures are very hard to interpret. Many English errors are present; some are listed in the Section below. Several numbers in the Abstract seem redundant.A major concern is the sample size as the authors recognize. Also why were only 18/44 subjects evaluated at the beginning? The discussion is very long;  perhaps the comparisons of circulating cells with other parameters ( albumin etc )might deserve a separate paper. (These results are quite interesting)

Comments on the Quality of English Language

Extensive editing is needed. Examples of this are Ines 51,, 67,  77, 78 and many others.

Reviewer 3 Report

Comments and Suggestions for Authors

Ilaria et al demonstrated that the levels of CMMCs in MM were associated with serum albumin, calcium and macroglobulin. The levels of CMMCs in MM are related to patients’ responses to the treatment.  Data from the present study provide a good strategy to evaluate the MM treatment. However, there are some more detailed data needed for publication.

1.        It would be better to use flow chart to describe how CMMC works. CMMCs should be DAPI- cells. It needs to be clarified whether CD138 antibody binds to normal cells as well.

2.        Authors characterized BM samples using various antibodies and flow cytometry. However, I could not any flow data and flow chart in the present manuscript.

3.        Authors divided patients into coMMstant=1 and coMMstatn=0 patients. It would be better to describe how those patients separated and why. In Figure 5B, there are similar CMMCs in both coMMstant=1 and coMMstatn=0 patients, particular in induction and pre-maintenannce phases. How do authors explain it?

Reviewer 4 Report

Comments and Suggestions for Authors

Very well prepared and detailed manuscript. The authors present the results obtained using the chosen methodology in an orderly and logical manner. They present the conclusions in a critical and thoughtful manner. The selection of tables and figures allows for tracing the discussed data. My only comment is a request to present the limitations of the research in a separate paragraph Limitations.

Round 2

Reviewer 2 Report

Comments and Suggestions for Authors

Comments and corrections are appropriate.

Reviewer 3 Report

Comments and Suggestions for Authors

The Authors have addressed all of my concerns with the original manuscript. The paper will make a good contribution to the field.